# Changes in Saliva Analytes Associated with Lameness in Cows: A Pilot Study

**DOI:** 10.3390/ani10112078

**Published:** 2020-11-09

**Authors:** María D. Contreras-Aguilar, Pedro Javier Vallejo-Mateo, Rasa Želvytė, Fernando Tecles, Camila Peres Rubio

**Affiliations:** 1Interdisciplinary Laboratory of Clinical Analysis of the University of Murcia (Interlab-UMU), Veterinary School, Campus Mare Nostrum, University of Murcia, 30100 Murcia, Spain; mariadolores.contreras@hotmail.com (M.D.C.-A.); camila.peres@um.es (C.P.R.); 2Department of Animal Medicine and Surgery, Veterinary School, Campus Mare Nostrum, University of Murcia, 30100 Murcia, Spain; pedroja512@hotmail.com; 3Department of Anatomy and Physiology, Research Center of Digestive Physiology and Pathology, Veterinary Academy, Lithuanian University of Health Sciences, Tilzes Str. 18, LT-47181 Kaunas, Lithuania; Rasa.Zelvyte@lsmuni.lt

**Keywords:** biomarkers, dairy cows, lameness, saliva, sialochemistry

## Abstract

**Simple Summary:**

Saliva may contain useful biomarkers which provide information about animal welfare using convenient and non-invasive sampling methods. In addition, the development of automated techniques of measuring analytes in saliva provides advantages from the technical point of view since they are cost-effective, reliable, and replicable. In this study, 21 salivary analytes measured by automated assays were tested as potential biomarkers of lameness, one of the most prevalent diseases in dairy cows producing significant economic losses. As a result, total esterase (TEA) showed increases in saliva in a group of 11 cows with lameness, decreasing when the lameness was solved after a specific treatment consisting of a hoof trimming and a medical treatment. In addition, TEA activity correlated with the severity of the lameness. Further studies using a larger population of cows with different causes of lameness and severity should be performed to determine the potential of TEA as a biomarker of lameness in cows.

**Abstract:**

The possible changes in a panel of 21 salivary analytes on a population of cows with lameness before and after treating lameness by hoof trimming were analyzed. Then, the analytes that showed significant changes were studied in a larger population of cows with lameness and compared with healthy cows For this purpose, two groups of cows were made by a specialized veterinarian. One consisted of healthy cows with no external signs of diseases and no hematological or biochemical abnormalities, and showing no signs of lameness according to the numerical rating system of severity (NRS, 5-point scale); and the other composed of cows showing only lameness with a NRS of 3.1 ± 0.87 and a lesion scoring system (LSS, 4-point scale) of 3.3 ± 0.89. Both groups did not differ in parity (*p* = 0.140), days in milk (DIM) (*p* = 0.780), and body condition score (BCS) (*p* = 0.074). Initially, 21 biochemical analytes were determined in the saliva of six cows with lameness at the diagnosis time (T0) and twenty days after hoof trimming that successfully solved the lameness (TF). This exploratory study only showed significantly higher values in lipase (Lip) and total esterase (TEA) at T0 compared to TF (*p* < 0.001 and *p* = 0.034, respectively). When both analytes were measured in the additional five lame cows and the results of all the animals of the lame group (n = 11) were compared with the healthy group (n = 11), only TEA showed higher activities in the group of lame cows than healthy cows (*p* = 0.004). TEA was positively correlated with both NRS and LSS (r = 0.43, *p* = 0.004 and r = 0.35, *p* = 0.003). In conclusion, this study showed that cows with lameness in our experimental conditions had higher TEA values than healthy cows, and these values decreased after treatment. This is a pilot study, and further studies using a larger population of cows with lameness due to different causes and severity should be performed to determine the potential of TEA as a biomarker of lameness in cows.

## 1. Introduction

Lameness is, together with infertility and mastitis, one of the most prevalent diseases in dairy herds producing significant economic losses, reporting mean prevalence of lameness in dairy cows to be 31.6% in UK [1] and 21% in three Canadian provinces [2]. It has been associated with reduced milk yield, decreased reproductive performance, and increased culling rates, producing significant economic losses [3,4].

In addition to the economic effect, lameness is recognized as a major welfare problem in cows. Cortisol has been extensively employed as a stress biomarker in lame cattle since acute pain induces activation of the hypothalamus-pituitary-adrenal (HPA) axis [5,6,7], although there is a controversy in the results provided in different reports. In an experimental oligofructose-induced lameness in dairy heifers, increases in serum cortisol and norepinephrine were found [7]. However, in a different study, no significant differences were detected in saliva and hair cortisol between feedlot cattle with and without lameness [8]. This controversy can be due to the high inter-individual variability in baseline cortisol concentrations since those can be affected by other factors different from stress, so it would be necessary to compare animals’ values relative to themselves and not to others.

Saliva rather than blood has been rising in interest in recent years as biologic fluid for the evaluation of welfare, stress, and disease in animals, since saliva can be easily and repeatedly collected in a large number of samples even at short-time intervals in a non-invasive way, without producing pain, and no need for specialized staff [9,10]. In this line, the measurement in the saliva of the same analytes that are usually included in a biochemistry profile performed in serum/plasma (“sialochemistry”) [11,12] allows exploring a relatively high number of analytes that could be of interest to evaluate stress, poor welfare, or disease conditions, as already done in horses [13]. However, to the author’s best knowledge, no study using saliva to evaluate changes in a panel of biochemical analytes in cows with lameness has been previously published. Only one study was found where a panel of salivary biomarkers was applied in cows. In this report, dairy cows with acute mastitis showed increases in alpha-amylase activity, cortisol, lactate, and uric acid, as well as decreases in cholinesterase levels in saliva [14].

The study hypothesizes that lameness in cows can produce changes in selected analytes in saliva and that possibly some of these analytes could reflect improvements in lameness after treatment. Therefore, this study aimed to evaluate changes in a panel of 21 salivary analytes in cows with lameness exploring the changes in these analytes after the lameness was solved.

## 2. Materials and Methods

Twenty-four Holstein-Friesian dairy cows from a commercial dairy herd located in the southeast of Spain (38°2′ N, 1°15′ W) were initially selected for this study. Information about the parity, days in milk (DIM), body condition score (BCS), and milk yield is described in Table 1. The BCS was categorized according to a visual scoring technique with a five-point scale (1 = emaciated to 5 = severely over-conditioned) [15]. Milk yield was measured by an automatic milking system (Westfalia Warragul, Yarragon, VIC, Australia), and each cow activity was monitored by sensors (Afimilk, Kibbutz Afikim, Israel).

The cows were fed with a total mixed base-ration offered once daily ad libitum at 08:30. From −50 days relative to calving, they received a far-off diet (1.49 Mcal/kg of dry matter, 9.4% rumen-degradable protein, and 5.3% rumen-undegradable protein), and once calving the cows received a lactation diet (1.71 Mcal/kg of dry matter, 11.0% rumen-degradable protein, and 6.0% rumen-undegradable protein). Water intake was available ad libitum. All the cows included in this study were housed in free-stalls (1.1 stalls/cow) but with straw, and they were checked daily and milked two times a day.

Two groups of cows were made. The lameness group (L group) was initially composed of twelve cows identified by a specialist veterinarian (PJV-M) for showing signs of lameness using a numerical rating system (NRS) based on a 5-point scale, in which a score of 1 represented a no-lame animal and 5 represented a severely lame animal (coefficient of determination, intra-class > 0.75, inter-class > 0.68) [16]. The other group was also initially integrated by twelve healthy cows (H group) selected for having similar parity and productive period than the L group (Table 1) and scored with value 1 by the NRS. Cows were routinely checked by a veterinarian (PJV-M) during the experimental study, and animals showing other health issues different than lameness (e.g., mastitis, metritis, or ketosis) in the L group, or any of them in the H group or hematological or biochemical abnormalities along the period of study, were consequently removed from the study. Consequently, two cows from L and H group had to be removed from the study since one died during the study, and the other developed diarrhea episodes, respectively. Therefore, 22 cows (11 from L group and 11 from H group) were finally evaluated. The Bioethical Committee (“Comité Ético de Experimentación Animal”, CEEA) from Murcia University (Spain) with the number 171/2015 approved this study.

Once lameness was detected in the L group, one day later the cows were subjected to hoof trimming to characterize and treat the lameness. At that moment, two hoof-trimmers trained examined each hoof and recorded the presence of lesions by a 4-point lesion scoring system (LSS) according to Flower et al. [16] (1 = slight discoloration; 2 = moderate hemorrhagic lesion; 3 = severe hemorrhagic lesion; 4 = sole ulcer, exposed corium), and other possible foot pathologies (e.g., interdigital necrobacillosis, osteomyelitis or active digital dermatitis, among others) [4]. In addition to the hoof trimming, a conventional medical treatment based on tylosin (10,000–20,000 IU/Kg/day) for five days or ceftiofur (1 mg/Kg/day) for three days combining with ketoprofen (3 mg/Kg/day) for three days was administered. Lameness states were again scored just before the hoof trimming treatment by the two hoof-trimmers trained, being the values used for characterizing the lameness states in the L group.

The sampling time T0 in the L group was performed the day of diagnosis before starting the hoof trimming treatment. Twenty days after T0, when all the L group cows showed a NRS of 1, samplings were again done (TF). Otherwise, the H group was sampling one day after the L group was sampling at T0 and TF, respectively. The samplings were made in the milking parlor right after the milking, after removing the milkers, and while the nipples were post-dipped, at the moment when the sampler had the opportunity to do it. The sampling procedure was always performed in the first milking between 07:00 and 12:00. All cows were sampled in a similar period of the day at each sampling time, with differences lower than two hours.

First, saliva samples were collected by introducing a sponge clipped to a flexible thin metal rod into the cow’s mouth to be chewed. The sponge was then placed in collection devices (Salivette, Sarstedt, Aktiengesellschaft & Co, Nümbrecht, Germany). Just after saliva sampling, blood was obtained by venipuncture in vena caudal using a sodium heparin device (LH/Li Heparin, Aquisel, Barcelona, Spain). Both saliva and blood devices were stored in ice until arrival at the processing laboratory (less than 2 h), where they were then centrifuged at 3.000× *g* for 10 min at 4 °C. Saliva and plasma specimens were stored at −80°C (less than five months) until analysis.

The salivary analytes evaluated in the chemistry profile were:Enzymes: aspartate aminotransferase (AST), alanine aminotransferase (ALP), γ-glutamyl transferase (gGT), lipase (Lip), alpha-amylase, lactate dehydrogenase (LDH), creatine kinase (CK), butyrylcholinesterase (BChE), total esterase (TEA), and adenosine deaminase (ADA).Metabolites: creatinine, urea, uric acid, total bilirubin, triglycerides, glucose, and lactate.Proteins: total proteins and albumin.Minerals: phosphorus and total calcium.Hormones: cortisol.

This chemistry profile was carried out on an automated chemistry analyzer (Olympus Diagnostica GmbH AU 600, Beckman Coulter, Ennis, Ireland) and adapted to saliva, as previously reported [13]. Salivary cortisol (sCor) was analyzed using a chemiluminescent immunoassay system (Immulite 1000, Siemens Healthcare Diagnostic, Deerfield, IL, USA), according to Escribano et al. [17]. Amylase, BChE, ADA, uric acid, lactate, and cortisol were previously evaluated and validated in cow’s saliva [12]. An analytical validation of the remain analytes included in this study was made, and all showed an inter- and intra-variability lower than 10% of the coefficient of variation (CV), being linear when applied to serial sample dilutions (R2 = 0.997 ± 0.004) (Appendix A). Total bilirubin could not be detected in saliva with the assays employed; therefore, this analyte was not evaluated in this study. Cortisol in plasma (pCor) was analyzed as described previously in saliva.

In our study, the salivary chemistry profile was applied in an exploratory population of six cows from the L group before (T0) and after (TF) the treatment selected randomly to evaluate possible differences in the salivary chemistry profile. Then, the salivary analytes that showed significant changes and pCor were measured in the additional 5 lame cows (n = 11) and the healthy group (n = 11).

Previous the statistical analysis, data were checked for normality using the Shapiro-Wilk test, and those showing no normality were previously log-transformed to normalize it. The parity, DIM, and BCS at T0 in both groups were evaluated by an unpaired Student’s t-test (2-tailed) to guarantee no differences between groups. No differences in parity (t21 = 1.53, *p* = 0.140) and DIM (t21 = 0.27, *p* = 0.780) were observed, although a tendency was in BCS (t21 = 1.90, *p* = 0.074). Therefore, a mixed linear model study was performed using the BCS as a covariate in the salivary analytes’ results previously selected after the exploratory study. An unpaired Student’s t-test (2-tailed) was also performed in the milk yield at T0 in both groups to assess possible differences.

For salivary analytes evaluation, a paired Student’s t-test (2-tailed) was performed to assess the possible significant differences between T0 and TF in the exploratory population (six lame cows). In those analytes that showed significant changes, a stand-alone power program for statistical testing (G-Power) [18] was employed using their means and standard deviations to establish the minimum number of individuals that would be necessary to be included in the global population for reaching a significance level of α = 5% (*p* < 0.05) and a power of 80%. It showed that a minimum number of 10 animals per group would be needed to get appropriate results for the selected analytes at different times. The two-way ANOVA of repeated measures followed by Sidak’s multiple comparisons tests were then used to assess whether significant changes existed between the 11 cows from the L group and the 11 cows from the H group.

Spearman or Pearson r, depending on whether the data were non-normally or normally distributed, were calculated in the selected salivary analytes in the global population to evaluate the correlation between them and the pCor concentrations, NRS, and LSS. An r-value between 0.90 to 1 was considered to have a very high correlation, 0.70 to 0.90 high correlation, 0.50 to 0.70 moderate correlation, 0.30 to 0.50 low correlation, and less than 0.30 little if any correlation [19].

All statistical analyses were performed using the commercial statistics package GraphPad Prism 8 (GraphPad Software Inc., La Jolla, CA, USA). Values of *p* ≤ 0.05 were selected to indicate significance in all analyses.

## 3. Results

### 3.1. Characteristics of the Cows of the Study

Lower milk yield was observed in the L group compared to H group (1.30 fold, t21 = 2.35, *p* = 0.028).

L Group showed a NRS of 3.1 ± 0.87 and a LSS of 3.3 ± 0.89 on the day of lameness diagnosis. All cows showed lameness processes associated with non-infectious lameness (ulcers, white line disease/abscesses, and toe ulcers).

Cortisol in plasma showed changes between groups (F1,20 = 10.49, *p* = 0.014), with higher concentrations in the L group (2.41 ± 1.09 µg/dL) than H group (1.30 ± 0.86 µg/dL) at T0 (*p* = 0.05). No significant differences between L and H groups at TF (1.40 ± 1.34 µg/dL and 0.70 ± 0.37 µg/dL, respectively) were observed (*p* = 0.45).

### 3.2. Differences in the Salivary Chemistry Profile before and after Lameness Treatment

When salivary analytes were evaluated in the exploratory population before and after lameness treatment (Table 2), only Lip and TEA showed changes between the time of diagnosis (T0) and when the lameness lesions were resolved (TF), with higher activities at T0 than at TF (2.6-fold, *p* < 0.001; 1.5-fold, *p* = 0.034, respectively).

### 3.3. Changes in the Salivary Analytes between Lameness and Healthy Cows

Lip (time effect, F1,20 = 10.68, *p* = 0.004) and TEA (time × group effect, F1,20 = 4.65, *p* = 0.044) showed no change between times in the H group (*p* = 0.501 and *p* = 0.714, respectively), but higher activities at T0 than TF (1.6-folds, *p* = 0.004; 1.4-folds, *p* = 0.002, respectively) in the L group (Figure 1). Higher activities of TEA were observed in L group compared with H group at T0 (1.5-fold, *p* = 0.050). The mix linear model study showed no significant effect of the BCS in Lip (*p* = 0.728) and TEA (*p* = 0.429) results.

### 3.4. Correlations between Parameters

Correlations results are showed in Table 3. Lip and TEA correlated moderately (r = 0.58, 95% confidence interval (CI) 0.34–0.75, *p* < 0.001) between them. TEA activity showed a low positive correlation with the NRS (r = 0.43, 95% CI 0.14–0.65, *p* = 0.004) and the LSS (r = 0.35, 95% CI 0.04–0.60, *p* = 0.003), while Lip only with the NRS (r = 0.33, 95% CI 0.02–0.57, *p* = 0.031). No correlations were observed between TEA (*p* = 0.723) and Lip (*p* = 0.936) in saliva with pCor.

## 4. Discussion

In our study, changes in salivary analytes in cows with lameness were evaluated. The lameness severity showed a scoring mean of 3, considered as moderate [20]. This degree of severity represents more the usual situation on the farms of our area where more severe lameness are not so frequent. Although moderate, the lameness in our study was associated with lower milk yield and a higher mean pCor. These facts would indicate that the lame produce harmful effects and reduced welfare in the cows, as previously described [21,22]. Distribution of parity, BCSs, and DIM did not differ between the lame and healthy groups, so there was no underlying productivity/parity cause that could interfere with the results of our study [23].

In this trial, an extensive salivary chemistry profile was initially applied in lame cows before and after a treatment consisting of hoof-trimming. The evaluation of an extensive chemistry profile in saliva has been used in other species, such as horses [13], and it allows to select of those analytes that change in a particular condition. Only two analytes, Lip and TEA, from the 21 analytes evaluated showed changes in cows before and after lameness resolution. Both enzymes were significantly increased and correlated with the grade of lameness and/or the hoof lesion. However, when both analytes were evaluated in a larger population of lame and healthy animals, only TEA showed significant increases in cows with lameness. Previous studies reported significant increases in TEA and Lip in lame pigs showing pain and discomfort [24,25]. In pigs, the TEA activity in the saliva is integrated by the isoenzyme carbonic anhydrase IV (CA-IV), Lip, cholinesterase (ChE), and cholesterol esterase (CEL) activities, among others [24]. This would explain the moderate positive correlation between Lip and TEA observed in our study. Although Lip function in saliva is initially to degrade fats [26], it has been reported to increase after acute stress [26,27] due to sympathetic activation as occurs with the salivary alpha-amylase [28]. Since BChE did not show significant changes, the main influences in the TEA changes might be due to the isoenzyme CA-IV activity since it is the main component of TEA in saliva [29]. The salivary secretion of CA-IV also relates to the sympathetic nervous system (SNS) [30]. However, CA-IV increases could also be related to other physiological mechanisms, such as inflammatory conditions [24] or diseases, such as rumen acidosis, which secondarily could produce laminitis [31], since CA-IV secretion is also involved in the maintenance of the pH homeostasis [29]. Further studies should clarify the physio-pathological mechanisms of the increase in TEA activity in cows’ lameness.

No relationship was observed between TEA with pCor, probably because of the different release source, being TEA related to the SNS activation, while pCor related to the HPA axis. In a study made in horses, no correlation between an analyte related to SNS, such as sAA, and sCor when a painful disease was observed [32]. The different response dynamics could also contribute to this lack of correlation, since cortisol has a slower response after stress than catecholamines [7].

This study has various limitations. Although previous publications evaluating lame cows used a similar number of animals than in this report [7,33], it should be considered a pilot study due to the relatively low number of animals used and the inclusion of only dairy breeds. In addition, only lameness produced by non-infectious causes were included. Therefore, further studies involving a larger number of cows, different breeds, and evaluating different types and causes of lameness would be recommended to confirm these preliminary results. Another potential study design limitation was that the H group did not undergo a functional hoof trimming, and this did not allow us to evaluate whether a functional hoof trimming in health cows could potentially have any effect in the stress biomarkers.

In conclusion, the research hypothesis of our study was confirmed, and lameness in cows produced changes in selected analytes in saliva, that reflect improvements in lameness after treatment. Namely, cows with lameness in the conditions of this study showed increases in TEA activities in their saliva compared to healthy cows, which decreased when a specific hoof treatment was performed. This report should be considered a pilot study, and further studies in a large population with cows with lameness due to different causes and severity should be performed to determine the potential of TEA as a biomarker of lameness in cows. If the TEA is confirmed as a biomarker in these additional studies, point-of-care assays that allow the measurement of this analyte quickly and easily on-farm conditions could be developed and applied in routine by farmers or under the coordination of Dairy Herd Improvement associations.

## Figures and Tables

**Figure 1 animals-10-02078-f001:**
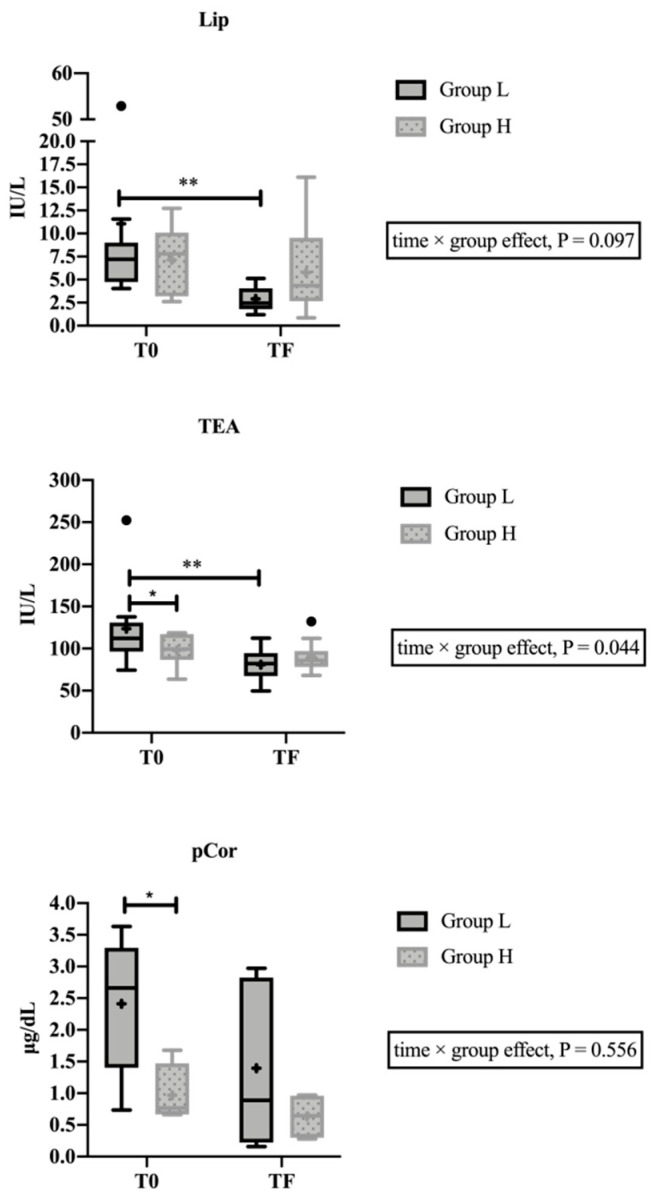
Results of lipase (Lip) and total esterase (TEA) in saliva, and cortisol (pCor) in plasma from a population of 22 cows divided into two groups, with lameness diagnosis (Group L, n = 11) and healthy cows (Group H, n = 11), sampled the day of diagnosis in group L or one day later in group H (T0), and twenty days after T0, when resolved their lameness lesions in group L or one day later in group H (TF). The plot shows median (line within box), 25th–75th percentiles (box), 5th and 95th percentiles (whiskers), and outliers (•). The cross inside the box shows the mean. Asterisk indicates statistically significant differences (Sidak’s multiple comparisons test) between times or groups (*: *p* < 0.05; **: *p* < 0.01).

**Table 1 animals-10-02078-t001:** Overview of the dairy cows enrolled in the lameness group (L Group) and the healthy group (H Group).

Productive Parameters	Descriptive Statistics	L Group	H Group
Parity	Mean ± SD ^1^	4.1 ± 1.04	3.5 ± 0.80
	95% CI ^2^	3.0–5.0	3.0–4.0
DIM ^3^ (days)	Mean ± SD	115 ± 65.7	106 ± 51.5
	95% CI	18–207	73–135
BCS ^4^ (score)	Mean ± SD	2.73–0.39	3.07 ± 0.31
	95% CI	2.50–3.00	2.75–3.50
Milk yield (kg/day)	Mean ± SD	31.3 ± 11.91	40.8 ± 6.97
	95% CI	20.7–42.5	36.0–45.0

^1^ SD, standard deviation; ^2^ CI, confidence interval; ^3^ DIM, days in milk; ^4^ BCS, body condition score.

**Table 2 animals-10-02078-t002:** Means (standard deviation) or medians [interquartile ranges, 25–75%] of a biochemical analysis performed in saliva from six dairy cows randomly selected from a larger population, the day of lameness diagnosis (T0), and after treatment when no sign of lameness was observed (TF). *p*-values in bold identify statistically significant differences between times.

Salivary Analytes	T0	TF	*p*-Value
AST ^1^ (IU/L)	9.5 [7.80–13.60]	12.6 [6.18–16.83]	0.172
ALP ^2^ (IU/L)	12.5 [7.93–27.80]	13.4 [6.63–19.65]	0.682
gGT ^3^ (IU/L)	41.2 (22.33)	28.4 (11.72)	0.532
Lip ^4^ (IU/L)	6.7 (1.56)	2.6 (1.22)	**<0.001**
sAA ^5^ (IU/L)	5.3 [2.25–3.95]	3.9 [3.25–7.95]	0.984
LDH ^6^ (IU/L)	48.2 [40.05–69.35]	36.5 [26.03–136.50]	0.973
CK ^7^ (IU/L)	2.9 (0.89)	4.1 (2.60)	0.403
BChE ^8^ (nmol/mL/min)	12.7 (6.34)	7.8 (3.68)	0.154
TEA ^9^ (IU/L)	111.4 (18.20)	74.0 (20.99)	**0.034**
ADA ^10^ (IU/L)	7.6 (4.23)	6.5 (4.66)	0.600
Creatinine (μmol/L)	22.1 (6.80)	25.6 (9.64)	0.367
Urea (mmol/L)	2.4 [2.15–2.67]	2.7 [1.86–3.29]	0.817
Uric acid (μmol/L)	10.1 (5.17)	7.7 (6.48)	0.521
Triglycerides (mmol/L)	0.14 [0.097–0.223]	0.08 [0.068–0.152]	0.113
Glucose (mmol/L)	0.01 [0.007–0.012]	0.02 [0.007–0.084]	0.226
Lactate (mmol/L)	0.50 [0.295–0.763]	0.25 [0.143–0.948]	0.765
Total protein (g/L)	0.78 [0.504–1.541]	0.82 [0.500–1.301]	0.839
Albumin (g/L)	0.3 (0.08)	0.3 (0.07)	0.695
Phosphorus (mmol/L)	6.4 (2.60)	6.5 (3.68)	0.937
Total calcium (mmol/L)	0.18 [0.127–0.407]	0.33 [0.249–0.893]	0.076
Cortisol (μg/dL)	0.190 (0.148)	0.172 (0.137)	0.836

^1^ AST, aspartate aminotransferase; ^2^ ALP, alkaline phosphatase; ^3^ gGT, γ-glutamyl transferase; ^4^ Lip, lipase; ^5^ sAA, salivary alpha-amylase; ^6^ LDH, lactate dehydrogenase; ^7^ CK, creatine kinase; ^8^ BChE, butyrylcholinesterase; ^9^ TEA, total esterase; ^10^ ADA, adenosine deaminase.

**Table 3 animals-10-02078-t003:** Correlation coefficients between Lipase (Lip) and total esterase (TEA) in saliva and plasma cortisol (pCor) concentrations, NRS, and LSS in 22 cows with lameness (n = 11) and healthy cows (n = 11).

Variable	Lip (IU/L)	TEA (IU/L)	PCor (µg/dL)
Lip ^1^ (IU/L)		**0.58 *****	−0.02
TEA ^2^ (IU/L)	**0.58 *****		0.09
pCor ^3^ (µg/dL)	−0.02	0.09	
NRS ^4^ (score)	**0.33 ***	**0.43 ****	**0.53 ***
LSS ^5^ (score)	0.28	**0.35 ***	**0.48 ***

^1^ Lip, lipase; ^2^ TEA, total esterase; ^3^ pCor, plasma cortisol; ^4^ NRS, numerical rating system of severity (5-point scale); ^5^ LSS, lesion scoring system (4-point scale). The correlation coefficients in bold showed significance (*: *p* < 0.05; **: *p* < 0.01, ***: *p* < 0.001).

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
