# Peer review of "Changes in Saliva Analytes Associated with Lameness in Cows: A Pilot Study"

_animals, 2020, doi:10.3390/ani10112078_

Round 1

Reviewer 1 Report

Interesting research paper. However major revision is required to adress the following:

Replace old references with recent ones. 

Add references in the Introduction section as related to the markers used for stress and production parameters.

Improve Results and Discussion sections. 

Add Conclusions section to clearly summarize your conclusions as a response to the research hypothesis. 

Author Response

Interesting research paper. However major revision is required to adress the following:

Point 1. Replace old references with recent ones.

Response 1. We have included in the new version of the manuscript some recent references such as:

  • Griffiths, B.E.; White, D.G.; Oikonomou, G. A cross-sectional study into the prevalence of dairy cattle lameness and associated herd-level risk factors in England and Wales. Front. Sci. 2018, 5, 1–8.
  • Solano, L.; Barkema, H.W.; Pajor, E.A.; Mason, S.; LeBlanc, S.J.; Zaffino Heyerhoff, J.C.; Nash, C.G.R.; Haley, D.B.; Vasseur, E.; Pellerin, D.; et al. Prevalence of lameness and associated risk factors in Canadian Holstein-Friesian cows housed in freestall barns. Dairy Sci. 2015, 98, 6978–6991.
  • Contreras-Aguilar, M.; Escribano, D.; Quiles, A.; López-Arjona, M.; Cerón, J.; Martínez-Subiela, S.; Hevia, M.L.; Tecles, F. Evaluation of new biomarkers of stress in saliva of sheep. Animal 2019, 13, 1278–1286
  • Contreras-Aguilar, M.D.; Escribano, D.; Martínez-Subiela, S.; Martínez-Miró, S.; Rubio, M.; Tvarijonaviciute, A.; Tecles, F.; Cerón, J.J. Influence of the way of reporting alpha- Amylase values in saliva in different naturalistic situations: A pilot study. PLoS One 2017, 12, 1–13.

If the reviewer thinks that we should add any additional reference of replace any reference, we would be happy to do it.

Point 2. Add references in the Introduction section as related to the markers used for stress and production parameters.

Response 2. Authors have included information (reference number 14) of a previous research in cows related to salivary biomarkers (lines 75-80): “However, to the author’s best knowledge, no study using saliva to evaluate changes in a panel of biochemical analytes in cows with lameness has been previously published. Only one study was found where a panel of salivary biomarkers in dairy cows with acute mastitis was evaluated, where increases in alpha-amylase activity, cortisol, lactate, and uric acid, and decreases in cholinesterase levels were observed [14].”

Point 3. Improve Results and Discussion sections.

Response 3. “Results” and “Discussion” sections were modified and improved according to the comments from reviewer 2.

Point 4. Add Conclusions section to clearly summarize your conclusions as a response to the research hypothesis. 

Response 4. We have included at the end a conclusion section where we summarize our conclusions as a response to the research hypothesis. This can be read as (lines 320-329): “In conclusion, the research hypothesis of our study was confirmed, and lameness in cows produced changes in selected analytes in saliva, that reflect improvements in lameness after treatment. Namely, cows with lameness in the conditions of this study showed increases in TEA activities in their saliva compared to healthy cows, which decreased when a specific hoof treatment was performed. This report should be considered a pilot study, and further studies in a large population with cows with lameness due to different causes and severity should be performed to determine the potential of TEA as a biomarker of lameness in cows. If the TEA is confirmed as a biomarker in these additional studies, point-of-care assays that allow the measurement of this analyte quickly and easily on-farm conditions could be developed and applied in routine by farmers or under the coordination of Dairy Herd Improvement associations.”

Reviewer 2 Report

General Comments:

The manuscript “Changes in saliva analytes associated with lameness in cow: a pilot study” provides interesting preliminary data to support the use of specific enzymes as biomarkers for identifying lameness. Although the paper does a good job in identifying the limitations and keeping their conclusions restricted to these results in this context as a pilot study, I also feel there are other limitations to the study design which need to be addressed. The main issue for me is how the groups were created (by a vet? Using what criteria) and the potential confounds which were not controlled for or addressed. If you are comparing lame cows to control cows between T0 and TF, both groups should have undergone hoof trimming and this information is not clear. Additionally, it was difficult to read this paper and understand what the authors were conveying; overall the English is quite good (and much better than my Spanish!), but it does need some modifications.

Specific comments:

L16-Perhaps you could change this to “Saliva may contain useful biomarkers which provide information about animal welfare using convenient and non-invasive methods.”

L19 and 27-You start by saying n=21 saliva samples and it is unclear how many are from lame vs. healthy cows and if there are replicate samples per cow. Additionally, later this is reduced to 12 cows in each group? But then actually 11 cows per group? Your sample size and experimental unit is the number of cows per group, not the number of saliva samples.

L29-define what is a “healthy” cow? No lameness or other disorders? Judged by who and how?

L31-37- why only look at 6 cows initially and more later? A pilot study to identify which values may change and if it is worth investigating with a follow up full study? Later you say this was all a pilot study.  Please explain.  (What you said in L141-145 is so much more clear!)

Were healthy vs. lame cows matched by parity or breed or what?

Follow up study – how many cows? How many saliva samples?

L36-37-Please change to “TEA was positively correlated with both NRS and LSS”

L45-even lameness alone is very prevalent and costly. Please provide some prevalence estimates of different countries.

L48-I don’t understand the relevance of reference #1

L49-55-please differentiate whether you are discussing the use of cortisol levels in general, or in response to something like a ACTH stimulation test. If the former, I think you need to briefly discuss the issues associated with making inferences based on cortisol alone (i.e. its correlation with activity in general and not just stress, the need for baseline measurements and comparing animals’ values relative to themselves and not to others, etc.)

L59-should read “without specialized staff” or “no need for specialized staff”

L59-63-entire sentence needs to be reworded to be clear

L62-Welfare can be poor or impaired, but not low.

L63-should read “to the author’s best knowledge”

L64-and no studies looking at lameness? If that is true, say so. And what did the mastitis study find?

L68-again where do the 21 samples come from? Which cows?

L71-now it says 24 cows?

L72-“years” meaning lactation number or parity? Or the actual age of the animal (which may not be so relevant)?

L76-What does “controlling each cow” mean?

L80-for the second diet, do you mean close-up or lactating cow diet? They can’t both be far-off diets.

L82-ab lib should be in italics

How many stalls available for how many cows?

L84-were not where

L86-88-should read “composed of” instead of “integrated by”

L91-Do you think it could be problematic that the lameness treatment is also confounded by body condition and milk yield?  Of course, low BCS and high milk yield are risk factors for lameness, but do you think there are difference in stress metabolites based on BCS or milk yield that you should have accounted for? Even to include as a covariate in your analyses.

L92-after you diagnosed lameness, you came up with a treatment protocol? That does not make sense

Did both lame and non-lame cows received hoof trimming?

L92-93- This description needs to be moved before or at least in the same paragraph where you describe the two groups of cows

L107-what does PJV stand for?  Who decided using which method which group the cows were put into (lame or not)?  You need to define lame vs. not and use objective criteria.

L109-should read “any of them” not “Whatever”

L113-remove the word “stop”.  Why did you sample after milking? Is there previous literature to support your methodology?

L123-“integrated”??

L134-138-Glad to see you have included which analytes were validated. What does “analytical validated mean”?  This does not make sense English-wise. Did you compare saliva to blood?

L147-“restore” should read “normalize”

L149-did you not test for parity differences? I would move the results of these t-tests to the methods to justify which you did not control for these variables which were different between groups (i.e. milk yield and potentially lactation? Even BCS in my opinion)

L151-155-What was the sample size determined by this calculation? Say this now, not in L169.

L162-Instead of referencing “the rule of thumb”, just explain what values of r mean what.

L170-172-Move this information to the methods section because your sample is 22 cows, not 24

L173- I have always considered 0.05 < P < 0.10 to be a tendency and therefore, the BCS does differ between groups. Also, what about lactation number?  Does the difference in milk yield not concern you? Either you need to justify why it is OK to be different between the groups, or you need to include it as a covariate in your analyses.

L176-you need to define what was a healthy/non-lame/sound cow in the methods and in the abstract.

L179-180-Were there any differences at TF between groups? If not, I think that is a good thing but please report this in the text, not just for the exploratory portion but also for the second part (it appears to be not different based on the figures).

L189- It looks like calcium tended to increase after treatment

L199-Figure 1 – make sure the axis all read T0 and TF, T1 is confusing and probably a mistake?

L220-223 – fix English please

L225, Table 1 and throughout manuscript – age?? Or parity

L2350243 - What is significance and function of lipase and esterase? Why does this make sense in terms of a lameness diagnosis and what is the mechanism? Please provide more references to literature, even if not specifically available for dairy cows.

L249-253 – this paragraph is good, but please add other limitations such as breed, management, or the experimental design which you think could be improved in the future.

Please briefly mention some implications and applications of your research - If saliva analytes were validated to be useful in identifying lameness, how do you recommend its use be implemented?  On-farm tests by farmers? Dairy Herd Improvement associations? Veterinarians? Explain.

L310 – this paper is from 1989, the reference is entirely wrong. Please correct and check other references.

Author Response

Response to Reviewer 2 Comments

Point 1. The manuscript “Changes in saliva analytes associated with lameness in cow: a pilot study” provides interesting preliminary data to support the use of specific enzymes as biomarkers for identifying lameness. Although the paper does a good job in identifying the limitations and keeping their conclusions restricted to these results in this context as a pilot study, I also feel there are other limitations to the study design which need to be addressed. The main issue for me is how the groups were created (by a vet? Using what criteria) and the potential confounds which were not controlled for or addressed. If you are comparing lame cows to control cows between T0 and TF, both groups should have undergone hoof trimming and this information is not clear.

Response 1. The authors would like to thanks the reviewer for his/her kind words. Regarding the creation of the group, both groups were created by a veterinarian (author’s acronym PJV-M) and using the NRS. This issue has been clarified in lines 103-114: “Two groups of cows were made. The lameness group (L group) was initially composed of twelve cows identified by a specialist veterinarian (PJV-M) for showing signs of lameness using a numerical rating system (NRS) based on a 5-point scale, in which a score of 1 represented a no-lame animal and 5 represented a severely lame animal (coefficient of determination, intra-class > 0.75, inter-class > 0.68) [16]. The other group was also initially integrated by twelve healthy cows (H group) selected for having similar parity and productive period than the L group (Table 1) and scored with value 1 by the NRS. Cows were routinely checked by a veterinarian (PJV-M) during the experimental study, and animals showing other health issues different than lameness (e.g., mastitis, metritis or ketosis) in the L group, or any of them in the H group or hematological or biochemical abnormalities along the period of study; were consequently removed from the study.”

Regarding the second question, the hoof trimming was only performed on the L group as part of their specific lame treatment. However, as the review indicated, this could be a potential study design confound. Therefore, it was added as an additional limitation in the new version of the manuscript (lines 316-319): “Another potential study design limitation was that the H group did not undergo a functional hoof trimming, and this did not allow us to evaluate whether a functional hoof trimming in health cows could potentially have any effect in the stress biomarkers.”

Point 2. Additionally, it was difficult to read this paper and understand what the authors were conveying; overall the English is quite good (and much better than my Spanish!), but it does need some modifications.

Response 2. We have made the modifications of English indicating by the reviewer, and also, we have applied the `Grammarly´ program to the text and made corrections of some grammar mistakes detected by the program.

Specific comments:

Point 3. L16-Perhaps you could change this to “Saliva may contain useful biomarkers which provide information about animal welfare using convenient and non-invasive methods.”

Response 3. The sentence has been changed according to the reviewer’s advice (lines 16-18).

Point 4. L19 and 27-You start by saying n=21 saliva samples and it is unclear how many are from lame vs. healthy cows and if there are replicate samples per cow. Additionally, later this is reduced to 12 cows in each group? But then actually 11 cows per group? Your sample size and experimental unit is the number of cows per group, not the number of saliva samples.

Response 4. We are sorry that we were not clear enough when explaining the sample number.

When the authors write “21 salivary analytes”, they want to refer to the analytes that were measured in the saliva from the six lame cows at T0 and TF to evaluate possible differences in salivary chemistry profile, not to the number of animals enrolled in this study.

Regarding our population (sample size or experimental unit), the initial population consisted of 12 cows identified for showing lameness, and 12 health cows with similar parity and productive period than the previous one. However, during the following 20 days after the T0 sampling, one cow from the lameness group was removed from our experiment because it died, and one cow from the healthy group was also removed since it showed diarrhea episodes when TF was collected. Therefore, finally, our sample size was of 11 cows/group.

We have tried to clarify this in the new version of the manuscript in:

  • Lines 20-25: “In this study, 21 salivary analytes measured by automated assays were tested as potential biomarkers of lameness, one of the most prevalent diseases in dairy cows producing significant economic losses. As a result, total esterase (TEA) showed increases in saliva in a group of 11 cows with lameness, decreasing when the lameness was solved after a specific treatment consisting of a hoof trimming and a medical treatment. In addition, TEA activity correlated with the severity of the lameness.”
  • Lines 87-88: “Twenty-four Holstein-Friesian dairy cows from a commercial dairy herd located in the southeast of Spain (38º2’ N, 1º15’W) were initially selected for this study”.
  • Lines 103-118: “Two groups of cows were made. The lameness group (L group) was initially composed of twelve cows identified by a specialist veterinarian (PJV-M) for showing signs of lameness using a numerical rating system (NRS) based on a 5-point scale, in which a score of 1 represented a no-lame animal and 5 represented a severely lame animal (coefficient of determination, intra-class > 0.75, inter-class > 0.68) [16]. The other group was also initially integrated by twelve healthy cows (H group) selected for having similar parity and productive period than the L group (Table 1) and scored with value 1 by the NRS. Cows were routinely checked by a veterinarian (PJV-M) during the experimental study, and animals showing other health issues different than lameness (e.g., mastitis, metritis or ketosis) in the L group, or any of them in the H group or hematological or biochemical abnormalities along the period of study; were consequently removed from the study. Consequently, two cows from L and H group had to be removed from the study since one died during the study and the other developed diarrhea episodes, respectively. Therefore, 22 cows (11 from L group and 11 from H group) were finally evaluated. The Bioethical Committee (`Comité Ético de Experimentación Animal´, CEEA) from Murcia University (Spain) with the number 171/2015 approved this study.”

Point 5. L29-define what is a “healthy” cow? No lameness or other disorders? Judged by who and how?

Response 5. The information required by the reviewer was added in the “Abstract” section (lines 30-35) and can be read as: “For this purpose, two groups of cows were made by a specialized veterinarian. One consisted of healthy cows with no external signs of diseases and no hematological or biochemical abnormalities, and showing no signs of lameness according to the numerical rating system of severity (NRS, 5-point scale); and the other composed of cows showing only lameness with a NRS of 3.1 ± 0.87 and a lesion scoring system (LSS, 4-point scale) of 3.3 ± 0.89.”

Point 6. L31-37- why only look at 6 cows initially and more later? A pilot study to identify which values may change and if it is worth investigating with a follow up full study? Later you say this was all a pilot study. Please explain.  (What you said in L141-145 is so much more clear!)

Point 8. Follow up study – how many cows? How many saliva samples?

Response 6 and 8. To make these two points more understandable, lines 40-44 were modified as in lines 186-189: “When both analytes were measured in the additional five lame cows (n = 11) and the results of all the animals of the lame group (n = 11) were compared with the healthy group (n = 11), only TEA showed higher activities in a group of lame cows than healthy cows (P = 0.004).”

Point 7. Were healthy vs. lame cows matched by parity or breed or what?

Response 7. The healthy cows were selected for having similar parity and productive period than the L group, and both factors did not differ between groups. This new information was added to the “Abstract” section (lines 35-37): “Both groups did not differ in parity (P = 0.140), days in milk (DIM) (P = 0.780), and body condition score (BCS) (P = 0.074).”

Point 9. L36-37-Please change to “TEA was positively correlated with both NRS and LSS”

Response 9. This change has been introduced in the new version of the manuscript (line 44-45).

Point 10. L45-even lameness alone is very prevalent and costly. Please provide some prevalence estimates of different countries.

Response 10. This new information has been added in the new version of the manuscript (lines 53-55): “Lameness is, together with infertility and mastitis, one of the most prevalent diseases in dairy herds producing significant economic losses, reporting mean prevalence of lameness in dairy cows to be 31.6% in UK [1] and 21% in three Canadian provinces [2].”

Point 11. L48-I don’t understand the relevance of reference #1

Response 11. This reference was replaced by the following: Weaver, M.D.; Jean, G.S.; Steiner, A. Bovine Surgery and Lameness, 2nd ed.; Blackwell: Oxford, UK, 2005, pp. 198–258 (lines 357-358).

Point 12. L49-55-please differentiate whether you are discussing the use of cortisol levels in general, or in response to something like a ACTH stimulation test. If the former, I think you need to briefly discuss the issues associated with making inferences based on cortisol alone (i.e. its correlation with activity in general and not just stress, the need for baseline measurements and comparing animals’ values relative to themselves and not to others, etc.)

Response 12. We have evaluated baseline cortisol level, not after any ACTH stimulation test. This issue in comparing only baseline cortisol levels was briefly discussed in the new version of the manuscript, as the reviewer well advised, in lines 64-66: “(…). This controversy can be due to the high inter-individual variability in baseline cortisol levels since those can be affected by other factors different from stress, so it would be necessary to compare animals’ values relative to themselves and not to others.”

Point 13. L59-should read “without specialized staff” or “no need for specialized staff”

Response 13. The correction was made (line 70).

Point 14. L59-63-entire sentence needs to be reworded to be clear

Response 14. The sentence was modified to make it more straightforward, and now it can be read in the following way (lines 70-75): “In this line, the measurement in the saliva of the same analytes included in a biochemistry profile performed in serum/plasma (“sialochemistry”)[11,12] allows exploring a relatively high number of analytes that could be of interest to evaluate stress, poor welfare or disease conditions, as already done in horses [13].”

Point 15. L62-Welfare can be poor or impaired, but not low.

Response 15. “Poor” welfare was finally written, removing the word “low” (line 74).

Point 16. L63-should read “to the author’s best knowledge”.

Response 16. The correction was added to the new version of the manuscript (line 75).

Point 17. L64-and no studies looking at lameness? If that is true, say so. And what did the mastitis study find?

Response 17. We have included that there are no studies using saliva to evaluate changes in a panel of biochemical analytes in cows with lameness has been previously reported.

Regarding the second question, the findings of the mastitis study have been included in the new version of the manuscript.

Overall, in the new version of the manuscript, it can be read (lines 75-80): “However, to the author’s best knowledge, no study using saliva to evaluate changes in a panel of biochemical analytes in cows with lameness has been previously published. Only one study was found where a panel of salivary biomarkers in dairy cows with acute mastitis was evaluated, where increases in alpha-amylase activity, cortisol, lactate, and uric acid, and decreases in cholinesterase levels were observed [14].”

Point 18. L68-again where do the 21 samples come from? Which cows?

Point 19. L71-now it says 24 cows?

Response 18 and 19. Again, we are sorry for this misunderstanding:

The sentences related to the number of cows included in this study (lines 87-88 and lines 103-116), were modified to solve the doubt and clarify: a) the number of analytes measured in saliva that are 21 analytes, and the number of animals used in the study that initially were 12 cows in the lame group and 12 cows in the healthy group, but that finally the sample size was 11 cows/group since two cows were removed from the study.

Point 20. L72-“years” meaning lactation number or parity? Or the actual age of the animal (which may not be so relevant)?

Response 20. “years” meant parity. So “years” was replaced by “parity” in all the new version of manuscript: line 89, 121, 191, and 281.

Point 21. L76-What does “controlling each cow” mean?

Response 21. The word “controlling” was removed, and now the sentence can be read in the following way (lines 92-93): “(…), and each cow activity is monitored by sensors (Afimilk, Kibbutz Afikim, Israel).”

Point 22. L80-for the second diet, do you mean close-up or lactating cow diet? They can’t both be far-off diets.

Response 22. The second diet is offered the lactating cows once they calved, and the first diet is only offered the cow once they were in – 50 d relative to calving. To make the sentences clearer, it was changes in the following way (lines 94-98): “From -50 d relative to calving, they received a far-off diet (1.49 Mcal/kg of dry matter, 9.4% rumen-degradable protein, and 5.3% rumen-undegradable protein), and once calving the cows received a lactation diet (1.71 Mcal/kg of dry matter, 11.0% rumen-degradable protein, and 6.0% rumen-undegradable protein).”

Point 23. L82-ab lib should be in italics.

Response 23. “ad libitum” in lines 94 and 98 were written in italics.

Point 24. How many stalls available for how many cows?

Response 24. This new information was added (lines 100-102): “All the cows included in this study were housed in free-stalls (1.1 stalls/cow) but with straw and they were checked daily and milked two times a day.”

Point 25. L84-were not where

Response 25. The word “where” was removed (line 101).

Point 26. L86-88-should read “composed of” instead of “integrated by”

Response 26. The reviewer advise was added in the new version of manuscript (line 104).

Point 27. L91-Do you think it could be problematic that the lameness treatment is also confounded by body condition and milk yield? Of course, low BCS and high milk yield are risk factors for lameness, but do you think there are difference in stress metabolites based on BCS or milk yield that you should have accounted for? Even to include as a covariate in your analyses.

Response 27. According to the reviewer’s concern, the BCS was used as covariate in a mixed linear model study. This has been included in the new version of the manuscript that can be read as (lines 191-198): “The parity, DIM, and BCS at T0 in both groups were evaluated by an unpaired Student’s t-test (2-tailed) to guarantee no differences between groups. No differences in parity (t21 = 1.53, P = 0.140) and DIM (t21 = 0.27, P = 0.780) were observed, although a tendency in BCS (t21 = 1.90, P = 0.074). Therefore, a mixed linear model study was performed using the BCS as a covariate in the salivary analytes’ results previously selected after the exploratory study. An unpaired Student’s t-test (2-tailed) was also performed in the milk yield at T0 in both groups to assess possible differences.” The results were the following (lines 247-253): “Lip (time effect, F1,20 = 10.68, P = 0.004) and TEA (time × group effect, F1,20 = 4.65, P = 0.044) showed no change between times in the H group (P = 0.501 and P = 0.714, respectively), but higher activities at T0 than TF (1.6-folds, P = 0.004; 1.4-folds, P = 0.002; respectively) in the L group (Figure 1). Higher activities of TEA were observed in L group compared with H group at T0 (1.5-fold, P = 0.050). The mixed linear model study showed no significant effect of the BCS in Lip (P = 0.728) and TEA (P = 0.429) results, also showing significant differences in the time effect in Lip (P = 0.049) and in time × group effect in TEA (P = 0.026) results.”

The milk yield was lower in the L group, probably due to the stressful situation due to the lameness, with higher levels of their plasma cortisol than in the healthy group. Correlation between milk yield and TEA or Lip was not observed (data not shown). If the reviewer considers it necessary to include this data in the new version of the manuscript, we could do it.

Point 28. L92-after you diagnosed lameness, you came up with a treatment protocol? That does not make sense.

Response 28. This paragraph was modified in order to make the methodology performed in this study more understandable (lines 123-133): “Once lameness was detected in the L group, one day later the cows were subjected to hoof trimming to characterize and treat the lameness. At that moment, two hoof-trimmers trained examined each hoof and recorded the presence of lesions by a 4-point lesion scoring system (LSS) according to Flower et al., [16] (1 = slight discoloration; 2 = moderate hemorrhagic lesion; 3 = severe hemorrhagic lesion; 4 = sole ulcer, exposed corium), and other possible foot pathologies (E.g., interdigital necrobacillosis, osteomyelitis or active digital dermatitis, among others) [4]. In addition to the hoof trimming, a conventional medical treatment based on tylosin (10 000 – 20 000 IU/Kg/day) during five days or ceftiofur (1 mg/Kg/day) during three days combining with ketoprofen (3 mg/Kg/day) during three days was administrated. Lameness states were again scored just before the hoof trimming treatment by the two hoof-trimmers trained, being these values used for characterizing the lameness states in the L group.”

Point 29. Did both lame and non-lame cows received hoof trimming?

Response 29. No, the hoof trimming was only performed on the L group as part of their specific lame treatment. This has been specified in the new version of the manuscript in lines 316-319 as limitation: “Another potential study design limitation was that the H group did not undergo a functional hoof trimming, and this did not allow us to evaluate whether a functional hoof trimming in health cows could potentially have any effect in the stress biomarkers.”

Point 30. L92-93- This description needs to be moved before or at least in the same paragraph where you describe the two groups of cows.

Response 30. The description was moved in the paragraph where the authors talk about the groups.

Point 31. L107-what does PJV stand for?  Who decided using which method which group the cows were put into (lame or not)?  You need to define lame vs. not and use objective criteria.

Response 31. PJV is acronymous of the author Pedro Javier Vallejo, a veterinary specialist in dairy cows. This new information is described in the paragraph from line 103 to 114.

Point 32. L109-should read “any of them” not “Whatever”

Response 32. The change was introduced in the new version of the manuscript (line 112).

Point 33. L113-remove the word “stop”.  Why did you sample after milking? Is there previous literature to support your methodology?

Response 33. The reason was only for practicality: when the milkers were removed to post-dip the nipples, the sampler took the opportunity to collect the cow saliva. This information was added to the manuscript (lines 153-155): “The samplings were obtained in the milking parlor right after the milking, after removing the milkers and while the nipples were post-dipped, moment when the sampler had opportunity to do it.” The word “stop” was removed (line 153).

Point 34. L123-“integrated”??

Response 34. The sentence was modified, and now it can be read as (line 165): “The salivary analytes evaluated in the chemistry profile were:”

Point 35. L134-138-Glad to see you have included which analytes were validated. What does “analytical validated mean”?  This does not make sense English-wise. Did you compare saliva to blood?

Response 35. The sentence was replaced by “An analytical validation of the remain analytes included in this study was made, (…)” (line 177-178). On the other hand, although it would have interest to do it, we did not compare results in saliva to blood in our study.

Point 36. L147-“restore” should read “normalize”

Response 36. This correction was added to the new version of the manuscript (line 191).

Point 37. L149-did you not test for parity differences? I would move the results of these t-tests to the methods to justify which you did not control for these variables which were different between groups (i.e. milk yield and potentially lactation? Even BCS in my opinion).

Response 37. Parity, together with the DIM and BCS, was included in the t-test study to corroborate no significant differences between groups. And these results were moved to the methods (lines 191-198): “The parity, DIM, and BCS at T0 in both groups were evaluated by an unpaired Student’s t-test (2-tailed) to guarantee no differences between groups. No differences in parity (t21 = 1.53, P = 0.140) and DIM (t21 = 0.27, P = 0.780) were observed, although a tendency in BCS (t21 = 1.90, P = 0.074). Therefore, a mixed linear model study was performed using the BCS as a covariate in the salivary analytes’ results previously selected after the exploratory study. An unpaired Student’s t-test (2-tailed) was also performed in the milk yield at T0 in both groups to assess possible differences.” The authors kindly invite the reviewer to see also response 27.

Point 38. L151-155-What was the sample size determined by this calculation? Say this now, not in L169.

Response 38. The sentence in lines 220-222 was removed and moved to the lines 200-206: “In those analytes that showed significant changes, a stand-alone power program for statistical testing (G-Power) [18] was employed using their means and standard deviations to establish the minimum number of individuals that would be necessary to be included in the global population for reaching a significance level of α =5% (P < 0.05) and a power of 80%. It showed that a minimum number of 10 animals per group would be needed to get appropriate results for the selected analytes at different times.”

Point 39. L162-Instead of referencing “the rule of thumb”, just explain what values of r mean what.

Response 39. The new information required by the reviewer was added in the new version of the manuscript (lines 212-219): “An r-value between 0.90 to 1 was considered to have a very high correlation, 0.70 to 0.90 high correlation, 0.50 to 0.70 moderate correlation, 0.30 to 0.50 low correlation, and less than 0.30 little if any correlation [19].”

Point 40. L170-172-Move this information to the methods section because your sample is 22 cows, not 24

Response 40. That information was moved to the paragraph found in lines 103-116.

Point 41. L173- I have always considered 0.05 < P < 0.10 to be a tendency and therefore, the BCS does differ between groups. Also, what about lactation number?  Does the difference in milk yield not concern you? Either you need to justify why it is OK to be different between the groups, or you need to include it as a covariate in your analyses.

Response 41. We understand the point of view of the reviewer. Therefore, due to the tendency in difference in BCS, the authors have made a mixed linear model study with the results obtained in Lip and TEA in healthy and lame cows at T0 and TF introducing the BCS as a covariate. As a result, BCS has not significant effect in Lip (P = 0.728) and TEA (P = 0.429) results. Therefore, this new information was added. The authors kindly invite the reviewer to see also response 27.

Regarding the difference between groups in the milk yield, the lower milk yield in the lameness cows compared to healthy cows has been previously described as consequence to the painful and stressful situation, with increases in plasma cortisol, as occurs in our experimental conditions. Therefore, the lower milk yield in lameness cows would be an effect of the lameness condition instead of the TEA or Lip results to be an effect of milk yield. In addition, correlation between milk yield and TEA or Lip was not observed (data not shown).

Point 42. L176-you need to define what was a healthy/non-lame/sound cow in the methods and in the abstract.

Response 42. It was defined:

  • In the “abstract” section (lines 30-33: “For this purpose, two groups of cows were made by a specialized veterinarian. One consisted of healthy cows with no external signs of diseases and no hematological or biochemical abnormalities, and showing no signs of lameness according to the numerical rating system of severity (NRS, 5-point scale); (…))
  • In the “Methods” section (lines 103-114: “The lameness group (L group) was initially composed of twelve cows identified by a specialist veterinarian (PJV-M) for showing signs of lameness using a numerical rating system (NRS) based on a 5-point scale, in which a score of 1 represented a no-lame animal and 5 represented a severely lame animal (coefficient of determination, intra-class > 0.75, inter-class > 0.68) [16]. The other group was also initially integrated by twelve healthy cows (H group) selected for having similar parity and productive period than the L group (Table 1) and scored with value 1 by the NRS. Cows were routinely checked by a veterinarian (PJV-M) during the experimental study, and animals showing other health issues different than lameness (e.g., mastitis, metritis or ketosis) in the L group, or any of them in the H group or hematological or biochemical abnormalities along the period of study; were consequently removed from the study.”

Point 43. L179-180-Were there any differences at TF between groups? If not, I think that is a good thing but please report this in the text, not just for the exploratory portion but also for the second part (it appears to be not different based on the figures).

Response 43. This new information was added in the new version of the manuscript (lines 232-234): “No significant differences between L and H groups at TF (1.40 ± 1.34 µg/dL and 0.70 ± 0.37 µg/dL, respectively) were observed (P = 0.45).”

Point 44. L189- It looks like calcium tended to increase after treatment.

Response 44. Yes, although only were selected those analytes which had differences after treatment with P ≤ 0.05.

Point 45. L199-Figure 1 – make sure the axis all read T0 and TF, T1 is confusing and probably a mistake?

Response 45. We thank the reviewer for this concern, because it was a mistake. The figure and the figure 1 legend were adequality modified (lines 253-254).

Point 46. L220-223 – fix English please

Response 46. The sentence was modified (lines 276-279): “This degree of severity represents more the usual situation on the farms of our area where more severe lameness are not so frequent. Although moderate, the lameness in our study was associated with lower milk yield and a higher mean pCor.”

Point 47. L225, Table 1 and throughout manuscript – age?? Or parity

Response 47. Parity. We modified it in all the manuscript.

Point 48. L2350243 - What is significance and function of lipase and esterase? Why does this make sense in terms of a lameness diagnosis and what is the mechanism? Please provide more references to literature, even if not specifically available for dairy cows.

Response 48. New information about the significance and function of lipase and esterase have been added in the new version of the manuscript (lines 292-305): “In pigs, the TEA activity in the saliva is integrated by the isoenzyme carbonic anhydrase IV (CA-IV), Lip, cholinesterase (ChE), and cholesterol esterase (CEL) activities, among others [24]. This would explain the moderate positive correlation between Lip and TEA observed in our study. Although Lip function in saliva (lingual lipase), which is directly secreted by the lingual glands, is initially to degrade fats [26], it has been reported to increase after acute stress [26,27] due to sympathetic activation as happening with the salivary alpha-amylase [28]. Since BChE did not show significant changes, the main influences in the TEA changes might be due to the isoenzyme CA-IV activity since it is the main component of TEA in saliva [29]. The salivary secretion of CA-VI also has a relationship with the sympathetic nervous system (SNS) [30]. However, its increases could also be related to other physiological mechanisms such as inflammatory conditions [24] or internal processes such as rumen acidosis, which secondarily could produce laminitis [31], since CA-IV secretion also acts in the maintenance of the pH homeostasis [29]. Therefore, further studies should clarify the physio-pathological mechanisms of the increase in TEA activity in cows’ lameness.”

Point 49. L249-253 – this paragraph is good, but please add other limitations such as breed, management, or the experimental design which you think could be improved in the future.

Response 49. This paragraph was modified according to the reviewer advises (lines 311-318): “This study has various limitations. Although previous studies evaluating lame cows used a similar number of animals than in this study [7,33], it should be considered a pilot study due to the relatively low number of animals used and the inclusion of only dairy breeds. In addition, only lameness produced by non-infectious causes were included. Therefore, further studies involving a larger number of cows, different breeds, and evaluating different types and causes of lameness would be recommended to confirm these preliminary results. Another potential study design limitation was that the H group did not undergo a functional hoof trimming, and this did not allow us to evaluate whether a functional hoof trimming in health cows could potentially have any effect in the stress biomarkers.”

Point 50. Please briefly mention some implications and applications of your research - If saliva analytes were validated to be useful in identifying lameness, how do you recommend its use be implemented?  On-farm tests by farmers? Dairy Herd Improvement associations? Veterinarians? Explain.

Response 50. We thank to the reviewer for the advice since it will enhance the quality of the manuscript. The following information has been added (lines 324-329): “This report should be considered a pilot study, and further studies in a large population with cows with lameness due to different causes and severity should be performed to determine the potential of TEA as a biomarker of lameness in cows. If the TEA is confirmed as a biomarker in these additional studies, point-of-care assays that allow the measurement of this analyte quickly and easily on-farm conditions could be developed and applied in routine by farmers or under the coordination of Dairy Herd Improvement associations.”

Point 51. L310 – this paper is from 1989, the reference is entirely wrong. Please correct and check other references.

Response 51. The mistake was resolved (line 390) and other references were checked.

REFERENCES

  1. Das, S.; Mitra, K.; Mandal, M. Sample size calculation: Basic principles. Indian J. Anaesth. 2016, 60, 652–656.

Round 2

Reviewer 1 Report

The revised manuscript now fulfills requirements for being published by Animals journal.